# Immuno-Molecular Targeted Therapy Use and Survival Benefit in Patients with Stage IVB Cervical Carcinoma in Commission on Cancer^®^-Accredited Facilities in the United States

**DOI:** 10.3390/cancers16051071

**Published:** 2024-03-06

**Authors:** Collin A. Sitler, Chunqiao Tian, Chad A. Hamilton, Michael T. Richardson, John K. Chan, Daniel S. Kapp, Charles A. Leath, Yovanni Casablanca, Christina Washington, Nicole P. Chappell, Ann H. Klopp, Craig D. Shriver, Christopher M. Tarney, Nicholas W. Bateman, Thomas P. Conrads, George Larry Maxwell, Neil T. Phippen, Kathleen M. Darcy

**Affiliations:** 1Gynecologic Cancer Center of Excellence, Department of Gynecologic Surgery and Obstetrics, Uniformed Services University of the Health Sciences, Walter Reed National Military Medical Center, Bethesda, MD 20889, USA; collin.a.sitler.mil@mail.mil (C.A.S.); tianc@whirc.org (C.T.); christopher.m.tarney.mil@health.mil (C.M.T.); batemann@whirc.org (N.W.B.); conrads@whirc.org (T.P.C.); george.maxwell@inova.org (G.L.M.); neil.phippen@usuhs.edu (N.T.P.); 2Murtha Cancer Center Research Program, Department of Surgery, Uniformed Services University of the Health Sciences, Walter Reed National Military Medical Center, Bethesda, MD 20889, USA; craig.shriver@usuhs.edu; 3The Henry M. Jackson Foundation for the Advancement of Military Medicine Inc., Bethesda, MD 20817, USA; 4Gynecologic Oncology Section, Women’s Services and The Ochsner Cancer Institute, Ochsner Health, New Orleans, LA 70115, USA; chad.hamilton@ochsner.org; 5Department of Obstetrics and Gynecology, Los Angeles School of Medicine, University of California, Los Angeles, CA 90024, USA; mtrichardson@mednet.ucla.edu; 6Palo Alto Medical Foundation, California Pacific Medical Center, Sutter Health, San Francisco, CA 94010, USA; chanjohn@sutterhealth.org; 7Department of Radiation Oncology, Stanford University School of Medicine, Stanford, CA 94305, USA; dskapp@stanford.edu; 8Division of Gynecologic Oncology, University of Alabama at Birmingham, O’Neal Comprehensive Cancer Center, Birmingham, AL 35249, USA; cleath@uabmc.edu; 9Gynecologic Oncology Division, Levine Cancer Institute, Atrium Health, Charlotte, NC 28204, USA; yovanni.casablanca@atriumhealth.org; 10Gynecologic Oncology Division, Stephenson Cancer Center, Oklahoma University Health Sciences Center, Oklahoma City, OK 73104, USA; christina-washington@ouhsc.edu; 11Gynecologic Oncology Division, GW Medical Faculty Associates, George Washington University, Washington, DC 20037, USA; nchappell@mfa.gwu.edu; 12Department of Radiation Oncology, University of Texas MD Anderson Cancer Center, Houston, TX 77030, USA; aklopp@mdanderson.org; 13Women’s Health Integrated Research Center, Women’s Service Line, Inova Health System, Falls Church, VA 22042, USA

**Keywords:** cervical cancer, survival, immunotherapy, chemotherapy, radiation, targeted therapy, stage IVB, external beam radiation, intracavitary brachytherapy, first-line treatment

## Abstract

**Simple Summary:**

Randomized clinical trials show a survival benefit associated with immuno-molecular therapy (IMT) use in metastatic or recurrent cervical cancer. This study investigated IMT use and survival in stage IVB cervical cancer patients in Commission on Cancer^®^ (CoC)^®^-accredited facilities. Patients diagnosed with stage IVB cervical cancer in the National Cancer Database and treated with first-line therapy with chemotherapy alone or with radiotherapy ± IMT were studied. Adjusted risks of death were estimated in patients treated with ±IMT after applying a propensity score analysis to balance the clinical covariates. There were 3164 evaluable patients, including 31% who were treated with IMT. The use of IMT increased from 11% in 2013 to 46% in 2019. In propensity-score-balanced patients, the median survival was 5 months longer with vs. without IMT. The adjusted risk of death was 28% lower following treatment with vs. without IMT. IMT was associated with a consistent survival benefit in real-world patients in (CoC)^®^-accredited facilities with stage IVB cervical cancer.

**Abstract:**

Purpose: To investigate IMT use and survival in real-world stage IVB cervical cancer patients outside randomized clinical trials. Methods: Patients diagnosed with stage IVB cervical cancer during 2013–2019 in the National Cancer Database and treated with chemotherapy (CT) ± external beam radiation (EBRT) ± intracavitary brachytherapy (ICBT) ± IMT were studied. The adjusted hazard ratio (AHR) and 95% confidence interval (CI) for risk of death were estimated in patients treated with vs. without IMT after applying propensity score analysis to balance the clinical covariates. Results: There were 3164 evaluable patients, including 969 (31%) who were treated with IMT. The use of IMT increased from 11% in 2013 to 46% in 2019. Age, insurance, facility type, sites of distant metastasis, and type of first-line treatment were independently associated with using IMT. In propensity-score-balanced patients, the median survival was 18.6 vs. 13.1 months for with vs. without IMT (*p* < 0.001). The AHR was 0.72 (95% CI = 0.64–0.80) for adding IMT overall, 0.72 for IMT + CT, 0.66 for IMT + CT + EBRT, and 0.69 for IMT + CT + EBRT + ICBT. IMT-associated survival improvements were suggested in all subgroups by age, race/ethnicity, comorbidity score, facility type, tumor grade, tumor size, and site of metastasis. Conclusions: IMT was associated with a consistent survival benefit in real-world patients with stage IVB cervical cancer.

## 1. Introduction

Stage IVB cervical cancer is rare, accounting for approximately 5% of all cervical cancer diagnoses. The American Joint Committee on Cancer (AJCC) and International Federation of Gynecology and Obstetrics (FIGO) define stage IVB cervical cancers as those extending beyond the pelvis with spread to distant lymph nodes ± organs [1,2]. In practice, treatment for this rare and diverse group of patients varies greatly, with a range of first-line treatment options, including chemotherapy (CT), radiotherapy (RT), immunotherapy, targeted therapy, or surgical resection [3]. First-line chemotherapy may include a combination of Cisplatin, Carboplatin, Topotecan, Paclitaxel, Bevacizumab, and/or Pembrolizumab [3]. RT may include external beam radiation (EBRT) alone or with intracavitary brachytherapy (ICBT) [3]. Surgery may or may not be used to treat cervical cancer. Despite these multiple therapeutic options, the prognosis for patients presenting with metastatic disease remains poor [4,5,6,7,8,9,10,11,12,13]. Definitive surgery is likely never a major consideration for patients with metastatic cervical cancer. Wang *et al.* [14] showed that the 5-year survival remained below 20% in the National Cancer Database (NCDB) for patients with stage IVB cervical cancer diagnosed between 2004 and 2015 and treated with either CT alone or CT + RT, with a median survival of 18.5 months for CT + EBRT vs. 27.5 months for CT + EBRT + ICBT.

Major advances in the treatment of patients with metastatic and recurrent cervical cancer have occurred over the last three decades. The transitions in the first-line treatment of advanced-stage cervical cancer, from Cisplatin monotherapy established in GOG-43 in 1985 [15] to combination Cisplatin-based therapy with Ifosfamide in GOG-110 in 1995 [16], paclitaxel in GOG-169 in 2004 [17], Topotecan in GOG-179 in 2005 [18], and Paclitaxel in GOG-204 in 2009 [19], have resulted in an extension in median survival from 7.1 months to 8.3, 9.7, 9.4, and 12.9 months, respectively. First-line therapy with Cisplatin and Paclitaxel ±EBRT ± ICBT was the most common standard-of-care treatment for metastatic cervical cancer prior to 2014 [20], with the survival advantage to be suggested based on the multimodality regimens [17,19,21,22]. In 2014, Bevacizumab, a monoclonal antibody targeting vascular endothelial growth factor (VEGF), was FDA-approved for use for metastatic and recurrent cervical cancer following the completion of the randomized phase III trial, GOG-240 [23]. The addition of Bevacizumab to CT for the treatment of advanced-stage cervical cancer further increased the median survival to 17.0 months [23,24].

Additional survival improvements have also been achieved with advances in radiation techniques, as well as the addition of immunotherapy and antibody drug conjugates [14,25,26,27,28,29,30,31,32,33,34,35,36,37,38]. This includes the randomized phase III BEATcc (GOG-3030/ENGOT cx10/GEICO 68-C/JGOG-1084) trial [26] showing the benefit of Atezolizumab with CT and Bevacizumab in metastatic and recurrent cervical cancer, with a median survival of 32.1 vs. 22.8 months. The KEYNOTE-826 trial [27] documented the survival benefit associated with Pembrolizumab plus CT ± bevacizumab in patients with metastatic, persistent, or recurrent cervical cancer, with a median survival of 24.4 months vs. 16.5 months. The Empower Cx/GOG-3016/ENGOT cx9 trial [37] reported the benefit of Cemilimab in recurrent cervical cancer, with a median survival of 12.0 vs. 8.5 months. The InnovTV 205/GOG-3024/EnGOT-CxS trial [36] demonstrated the manageable safety and encouraging antitumor activity of Tisotumab Vedotin plus Carboplatin, plus Pembrolizumab or plus Bevacizumab in metastatic or recurrent cervical cancer. The KEYNOTE A18/GOG-3047/ENGOT-cx11 trial is evaluating the addition of Pembrolizumab to CT and RT in locally advanced cervical cancer.

Musa *et al.* [39] recently reported on a trend in treatment patterns and costs, including the use of immune therapy and a relatively short duration of immune therapy in advanced cervical cancer. The study by Kim *et al.* [40] investigated funding decisions in Australia and the United Kingdon, focusing on the cost-effectiveness and financial risks associated with immunotherapy, which raised important considerations regarding the challenges in the equitable deployment of costly multimodality regimens in under-resourced settings and for patients with limited resources.

This study aimed to estimate the use of immuno-molecular therapy (IMT) with CT ± RT over the past decade and its impact on survival in patients with stage IVB cervical cancer in clinical care settings outside of RCTs. We utilized the NCDB, representing 70% of incident cancer diagnoses in the United States (U.S.) across 1500+ Commission on Cancer (COC)^®^-accredited facilities, to provide a hospital-based cancer registry view of trends to serve as a reference for future studies in non-COC^®^ facilities within the U.S., and in both developed and under-developed nations around the world. Assessments of treatment trends and survival outside RCTs are necessary as new standards of care are deployed to identify the factors that impact the treatment utilization and survival outcomes in clinical care facilities, which may merit further research or require extra resources and policy changes to achieve more equitable care and outcomes nationally and internationally for patients with metastatic diseases.

## 2. Methods

### 2.1. Study Design and Patient Selection Criteria

A retrospective cohort study was performed in patients who were diagnosed with stage IVB primary cervical cancer between 2013 and 2019 in the NCDB and who did not undergo surgery but did start first-line treatment within 60 days of diagnosis. The WCG Institutional Review Board made an exempt determination for this study under Protocol #14-1679, as these data were derived from a publicly accessible database with deidentified patient information. This study followed STROBE reporting guidelines.

Stage IVB was defined using the American Joint Committee on Cancer (AJCC)’s clinical staging system, seventh or eighth edition when available, or Collaborative Stage Site-Specific Factor 1 when the AJCC clinical stage was missing. Clinical covariates included age at diagnosis, race and ethnicity, year of diagnosis, comorbidity score, status of insurance, type of treatment facility, median neighborhood-derived income, geographic region, histologic subtype, tumor grade, tumor size, and distant site of metastasis as defined by the NCDB (see the footnotes for Table 1 for additional details).

IMT was defined in the NCDB as the first-course treatment, consisting of biological or chemical agents that alter the immune system or change the host’s response to tumor cells. These included immuno-molecular targeting agents such as immune checkpoint inhibitors and biologic response modifiers, and in 2013, the NCDB started recording antibody-based agents like bevacizumab as IMTs. The CT group included those patients who were treated with single-agent or multi-agent chemotherapy; patients who received an unspecified number of chemotherapy agents were excluded. The RT group included patients who were treated with external beam radiotherapy (EBRT) or external beam radiotherapy plus intracavitary brachytherapy (EBRT + ICBT); patients treated with radioisotopes or unspecified types of radiation were not included. EBRT represented EBRT delivered during phase I without any boost during a later phase. EBRT + ICBT referred to EBRT delivered during phase I and an ICBT boost at a later phase. Patients with hormonal therapy or other treatments were not included in this study. Patients with unknown status of IMT were excluded from the analysis. Patients for whom the initial treatment started beyond 60 days after diagnosis were also excluded.

### 2.2. Statistical Methods

All statistical analyses were conducted on SAS version 9.4 (SAS Institute, Cary, NC, USA) using two-sided tests, with significance set at *p* < 0.05. Adjustments were not made for multiple testing. Differences in demographic and clinical variables between the IMT and non-IMT groups were evaluated using *Chi*-square test (for categorical variables) or *t*-test (for age at diagnosis). A stepwise logistic regression analysis (stratified by year of diagnosis and geographic region) was performed to estimate the adjusted odds ratio (OR) and 95% confidence interval (CI) for clinical factors that were independently associated with the use of IMT.

Overall survival was calculated from the date of diagnosis to death or last contact. Survival distributions were estimated using Kaplan–Meier methods and compared using log-rank test. The relationship between the use of IMT and survival was initially evaluated using a multivariate Cox model, stratified by year of diagnosis and geographic region and adjusted for covariates including age at diagnosis, race/ethnicity, comorbidity score, status of insurance, type of treatment facility, median neighborhood-derived income, histologic subtype, tumor grade, tumor size, distant site of metastasis, and type of first-line therapy.

The association between the use of IMT and survival was further assessed after applying propensity score analysis. Propensity score analysis was performed as described by Kucera et al., with inverse probability of treatment weighting to balance the clinical covariates [41]. Specifically, a logistic model was applied to estimate each patient’s propensity to receive the IMT, conditional on age, race and ethnicity, year of diagnosis, comorbidity score, insurance status, facility type, neighborhood income, geographic region, histologic subtype, tumor grade, tumor size, metastatic site, and type of first-line therapy, with a patient with IMT assigned a weight of [1/propensity] and a patient without IMT assigned a weight of [1/(1 − propensity)] [42]. The quality of balance between the two groups was examined using the standardized mean difference, with a value of <10% considered to be well balanced [42].

The adjusted survival for IMT vs. non-IMT patients was estimated in the propensity-score-balanced cohort using the weighted Kaplan–Meier method, and adjusted hazard ratio (HR) for risk of death was estimated using a weighted multivariate Cox model, with 95% CI calculated using robust sandwich variance [43]. Adjusted survival for patients treated by [1] CT with vs. without IMT, [2] CT + EBRT with or without IMT, or [3] CT + EBRT + ICBT with or without IMT were estimated by reapplying the propensity score analysis, respectively. Subgroup analyses were also performed to explore whether the effect of IMT was consistent in different subgroups by age, race and ethnicity, comorbidity score, facility type, histologic subtype, tumor grade, tumor size, or metastatic site. Subgroup analyses were conducted using the original cohort by extending the multivariate Cox model with an interaction term.

## 3. Results

In total, there were 3164 patients with stage IVB cervical cancer who were diagnosed between 2013 and 2019 who met the study criteria (Figure 1). There were 836 patients who were excluded from this study, including 53 patients who were treated with hormonal or other first-line treatment, 117 patients who were treated with an unspecified number of chemotherapy agents, 4 patients with missing immuno-molecular therapy status, and 662 patients who started first-line treatment beyond 60 days after diagnosis.

Table 1 shows the clinical characteristics and first-line treatment that was provided to the patients with stage IVB cervical cancer. The mean age (standard deviation) was 54.2 (12.7) years old. The racial and ethnic distribution of the cohort was 64.2% non-Hispanic White, 17.7% non-Hispanic Black, 10.9% Hispanic, and 4% Asian and Pacific Islander patients. There were 19.3% of the patients who had a comorbidity score ≥ 1, 21.9% with Medicare, 26.6% with Medicaid, and 8.7% who were uninsured. Only 39% of patients were cared for in an Academic/Research Program, and 63.8% had a neighborhood-derived income < USD 63,333. The distribution by histology and disease spread included 64.7% of patients with squamous cell carcinoma, 16.1% with adenocarcinoma of the cervix, and 28.4% with both distant lymph node and organ metastases.

Of the first-line treatments utilized, 38.1% of patients were treated with CT alone, 42.7% with CT + EBRT, and 19.2% with CT + EBRT + ICBT. IMT was added to the first-line therapy for 969 (30.6%) patients. None of these patients were treated with surgery. The median follow-up was 49 months, and 2304 deaths had occurred at the time of analysis. The median survival time was estimated to be 14.9 months, with a survival rate of 58.8% at 1 year, 34.7% at 2 years, 26.3% at 3 years, and 19.8% at 5 years after diagnosis.

### 3.1. Factors Influencing Immuno-Molecular Therapy Use

The use of IMT increased over time from 10.5% in 2013 up to 46.4% in 2019 (Figure 2). Table 2 shows the clinical factors that were independently associated with the use of IMT in patients with stage IVB cervical cancer, including age, insurance status, facility type, sites of distant metastasis, and type of first-line treatment. Each 5-year increase in age at diagnosis was associated with a 10% reduction in the use of IMT (adjusted OR = 0.90, 95% CI = 0.85–0.95 *p* < 0.0001). Compared with patients with private insurance, those with Medicaid insurance were 26% less likely to receive IMT (adjusted OR = 0.74, 95% CI = 0.60–0.91, *p* = 0.005), whereas uninsured patients were 40% less likely to receive IMT (adjusted OR = 0.60, 95% CI = 0.43–0.83, *p* = 0.002). Patients treated at Non-Academic/Research Facilities were 25% less likely to receive IMT (adjusted OR = 0.75, 95% CI = 0.63–0.90, *p* = 0.002) than those at Academic/Research Facilities. Patients with both distant lymph node and organ metastases were more likely to receive IMT (adjusted OR = 1.27, 95% CI = 1.01–1.59, *p* = 0.042) than those with only distant lymph node disease. Patients treated with CT + EBRT were 39% less likely to be treated with IMT (adjusted OR = 0.61, 95% CI = 0.51–0.73, *p* < 0.0001), while those treated with CT + EBRT + ICBT were 75% less likely to be treated with IMT (adjusted OR = 0.25, 95% CI = 0.19–0.33, *p* < 0.0001) compared with patients who were treated with CT.

### 3.2. Relationship between Immuno-Molecular Therapy and Survival in Propensity-Score-Balanced Patients

Table 1 summarizes the demographic and clinical characteristics in patients who were treated with vs. without IMT in the original cohort and after applying propensity score analysis. The standardized mean difference for each of these clinical covariates was <10%, indicating that the covariates between the two treatment groups were well balanced.

The median survival time in propensity-score-balanced patients who were treated with IMT was 18.6 months, compared to 13.1 months for those who were not treated with IMT (Figure 3A). The addition of IMT to a first-line therapy with CT ± RT significantly improved survival, with a 28% reduction in the risk of death (adjusted HR = 0.72, 95% CI = 0.64–0.80, *p* < 0.0001).

The survival benefit associated with the addition of IMT to the first-line treatment was also suggested in all subgroups categorized by age, race and ethnicity, comorbidity score, facility type, tumor grade, tumor size, site of distant metastasis groups, and year of diagnosis (Figure 3B). Interestingly, the addition of IMT was associated with a significant survival improvement in patients with squamous cell carcinoma (adjusted HR = 0.68, 95% CI = 0.60–0.77, *p* < 0.0001), while patients with adenocarcinoma had a smaller non-significant survival improvement (adjusted HR = 0.81, 95% CI = 0.65–1.02).

The adjusted survival for IMT vs. no IMT patients was further analyzed according to the type of first-line therapy. The benefit of adding IMT to CT alone (adjusted HR = 0.72, 95% CI = 0.61–0.84; Figure 4A), CT + EBRT (adjusted R = 0.66, 95% CI = 0.56–0.78; Figure 4B), or CT + EBRT + ICBT (adjusted HR = 0.69, 95% CI = 0.45–1.04; Figure 4C) was generally consistent. The best survival was seen in the group of patients who received CT + EBRT + ICBT + IMT, with a survival rate of 52% at 5 years after diagnosis and a median survival time that was not reached as of the analysis.

## 4. Discussion

In this real-world study using hospital-based cancer registry data from the NCDB, we demonstrated the survival benefits associated with the addition of IMT to first-line treatments in patients with stage IVB cervical cancer. This survival benefit was demonstrated in patients treated with CT ± RT overall and within the subsets treated with CT, CT + EBRT, or the multimodality CT + EBRT + ICBT combination. The superior survival in stage IVB patients treated with vs. without IMT persisted after applying propensity score analysis to balance the clinical covariates that varied between these two treatment groups. Moreover, the survival benefit that was associated with the addition of IMT to the first-line treatment was suggested across all subgroups by age, race–ethnicity, comorbidity score, facility type, squamous histology, grade, and the site of metastasis.

The incremental survival benefit in patients with stage IVB with the addition of IMT after applying propensity score analysis extends to prior retrospective studies [5,10,16,20,21,23] and is consistent with results from phase II and phase III clinical trials [23,24,26,27,28,29,31,32,33,34,35,36,37,44,45]. Outside of clinical trials, Perkins et al. [25] reported a 24-month improvement in overall survival following CT + EBRT (whole pelvic radiation) vs. CT in a small multi-site retrospective cohort study with a median follow-up of 9 months. The study by Wang et al. [14] showed a 3.6-month improvement in median survival for stage IVB patients who were diagnosed between 2004 and 2015 and treated with CT + RT compared with CT alone, with 5-year survival rates reported to be <20% using the NCDB. Our study included patients who were diagnosed through 2019 and found that patients who were treated with CT + EBRT + ICBT ± IMT had a 5-year survival rate between 40 and 55%. This doubling in 5-year survival likely reflects improvements associated with the use of intensity-modulated RT, stereotactic body RT, and high-dose ICBT [14,20,21,22,30,46,47] and the addition of bevacizumab, pembrolizumab, antibody drug conjugates, biologic response modifiers, and immune checkpoint inhibitors in the treatment of metastatic, recurrent, and persistent cervical cancer [23,24,26,27,28,29,30,31,32,33,34,35,36,37,38,48]. The study by Musa et al. [39] showed that immunotherapy use has been increasing, but only a small subset of patients stayed on immunotherapy for prolonged periods, suggesting a need for more therapeutic options for first-line and second-plus-line treatments for metastatic, recurrent, or persistent cervical cancer.

This study also provides additional insights regarding the factors that impact the utilization of IMT in patients with stage IVB cervical cancer compared with prior studies [4,5,6,7,8,9,10,11,12,13,14,17,19,21,24,25,30,48,49,50,51,52,53]. We found that patients with Medicaid insurance or no insurance, older age, or either distant lymph node or distant organ metastasis but not with both, and those treated at Non-Academic/Research Facilities or with CT + EBRT ± ICBT were significantly less likely to receive IMT. Additional attention and research are required to study these structural barriers and develop strategies to mitigate these inequities. Pothuri et al. [54] are acknowledged for releasing the joint statement from the GOG Foundation and SGO regarding inclusion, diversity, equity, and access (IDEA) in gynecologic clinical trials. Efforts also need to extend to practices outside of clinical trials to ensure equity, access, and outcomes nationally and globally.

Our study was subject to the inherent limitations of a retrospective investigation of the NCDB, including missing or incomplete data [55]. In addition, the NCDB does not provide the opportunity to review or verify pathology or imaging reports or to access CT details regarding agents delivered (name and doses), particularly including specifics on which IMT was delivered. The IMT variable utilized by NCDB merges multiple classes of immuno-molecular targeting agents together, including immune checkpoint inhibitors, biologic response modifiers, molecular targeting agents like Bevacizumab, and antibody drug conjugates, making it impossible for us to evaluate the individual impact of these classes of agents independently. Our subset analysis in Figure 3B by year of diagnosis suggests a larger survival benefit trend following the addition of IMT in patients diagnosed between 2013 and 2015, 2016 and 2017, and 2018 and 2019, which is likely associated with the approved addition of bevacizumab in 2014 and, ultimately, immune checkpoint inhibitors and antibody drug conjugates starting in 2018 for the treatment of patients with metastatic and recurrent cervical cancer. The NCDB provided general characteristics and first-line treatment information for analysis but lacked additional details regarding the patient’s medical, screening, treatment, and surveillance histories to support more in-depth analyses. In addition, this study only focused on patients with stage IVB disease without surgery and did not include resected stage IVB patients or those with locally advanced or metastatic stage III or IVA disease, who also deserve attention. A lack of data regarding treatment for recurrent, persistent, or progressive disease were other limitations. In this study, we did not examine the effect of facility volume or evaluate the impact of intensity-modulated radiotherapy or three-dimensional planning at different facilities on outcomes, which are other opportunities for future investigations. With the change in cervical cancer staging in 2018, our data should be interpreted with caution, as patients were evaluated using two different staging criteria (~70% with seventh-edition and 30% with eight-edition AJCC stage criteria). Additional studies are needed to evaluate the benefit of IMT in squamous cell carcinoma vs. adenocarcinoma of the cervix to determine if the lack of benefit of IMT in adenocarcinoma was specific to those with stage IVB disease or more generalizable. We were also not able to correct for prior hysterectomy or classify patients by type of high-risk HPV infection, pathogenic alterations, PD-L1 staining, or other molecular features which may affect prognosis, nor did we have access to data regarding the progression-free survival or cause of death. The lack of patient-level income and education data and measures of structural determinants of health, smoking, diet, exposures, and nutrition were additional limitations of this study. Despite the large sample size of the NCDB and robust clinical data that are available to support retrospective hospital-based investigations, NCDB results may not be generalizable to the U.S. population or populations outside the U.S., as our data were derived exclusively from COC^®^-accredited facilities in the U.S. The NCDB also does not make all the clinical covariates associated with treatment assignment in stage IVB cervical cancer patients available for analysis [56]. Although the application of propensity score analysis to an observational cohort study provides more accurate survival estimates than conventional multivariate Cox modeling [42], it is not able to correct for bias associated with unmeasured variables, and our propensity scores may still result in an overestimation of treatment effects [56]. Despite these limitations, our study was strengthened by its large number of captured patients, with a geographically, racially, and ethnically diverse patient population, and the use of robust statistical methods that yielded results that are consistent with clinical trials [57].

## 5. Conclusions

The use of immuno-molecular targeted agents, labeled as IMT in the NCDB, increased steadily from 11% in 2013 to 46% in 2019 in stage IVB cervical cancer patients in COC^®^-accredited facilities. The addition of IMT appears to improve survival when added to the first-line treatment of stage IVB cervical cancer overall and in all subgroups, including those who were treated with CT, CT + EBRT, or CT + EBRT + ICBT, but not in patients with stage IVB adenocarcinoma.

## Figures and Tables

**Figure 1 cancers-16-01071-f001:**
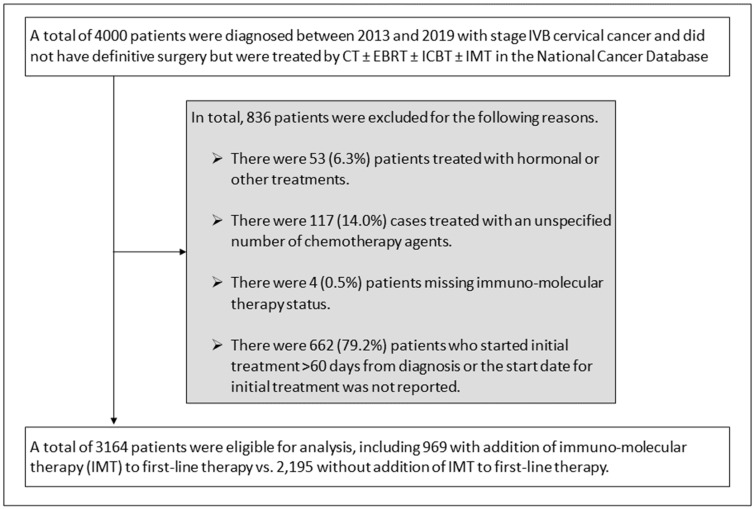
Schema for study selection and exclusions. Abbreviations: chemotherapy alone (CT); external beam radiotherapy (EBRT); intracavitary brachytherapy (ICBT); immuno-molecular therapy (IMT).

**Figure 2 cancers-16-01071-f002:**
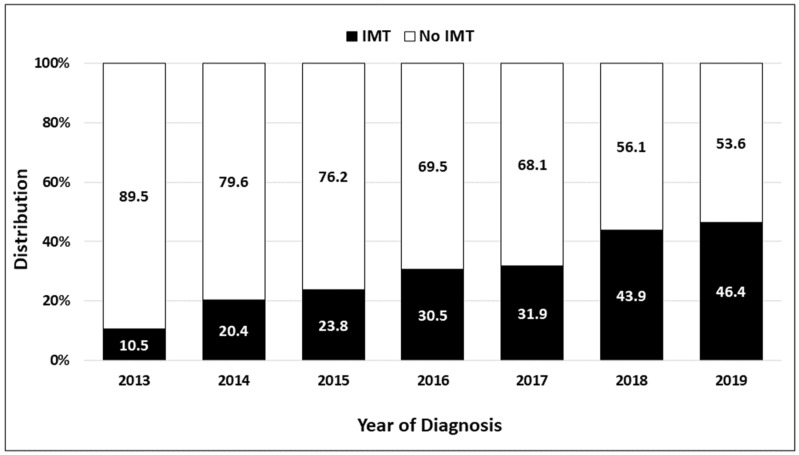
Use of immuno-molecular therapy (IMT) in patients diagnosed from 2013 to 2019 with stage IVB cervical cancer in the National Cancer Database.

**Figure 3 cancers-16-01071-f003:**
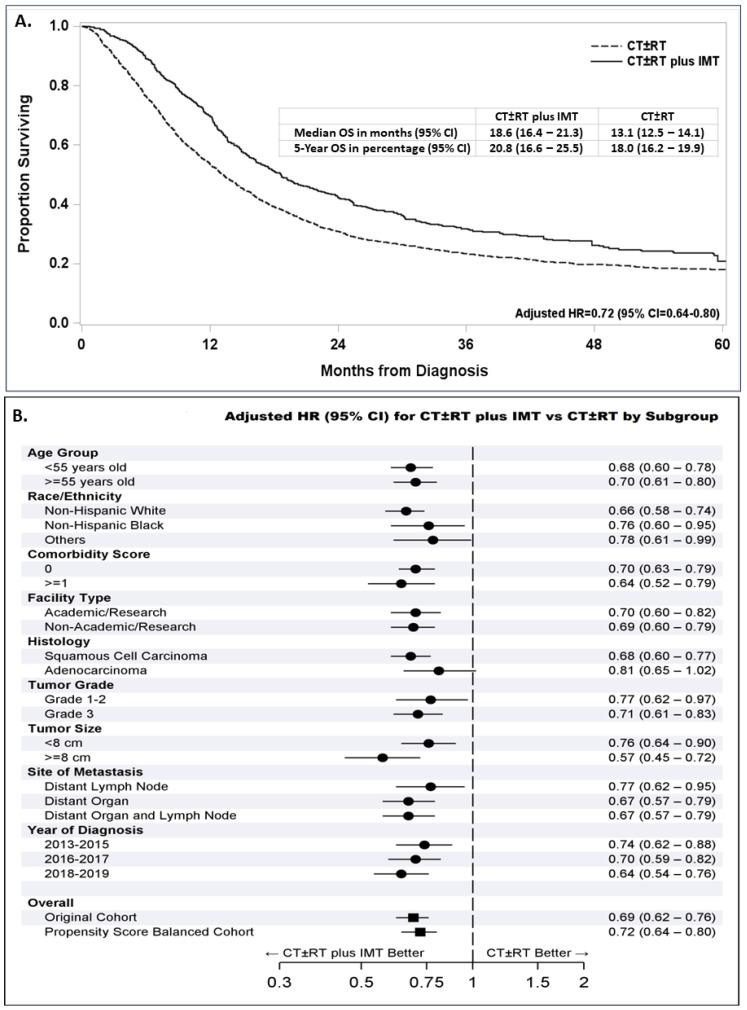
Survival following treatment with or without immuno-molecular therapy (IMT) added to chemotherapy alone or chemotherapy with or without radiation (CT ± RT) in propensity-score-balanced patients diagnosed between 2013 and 2019 with an embedded adjusted hazard ratio (AHR) and 95% confidence interval (CI) for patients treated with CT ± RT plus IMT compared with CT ± RT (**A**). Forest plot for all patients with stage IVB cervical cancer diagnosed between 2013 and 2019 and subgroups of these patients showing the reduced AHR (95% CI) following first-line treatment with CT ± RT plus IMT compared with CT ± RT (**B**).

**Figure 4 cancers-16-01071-f004:**
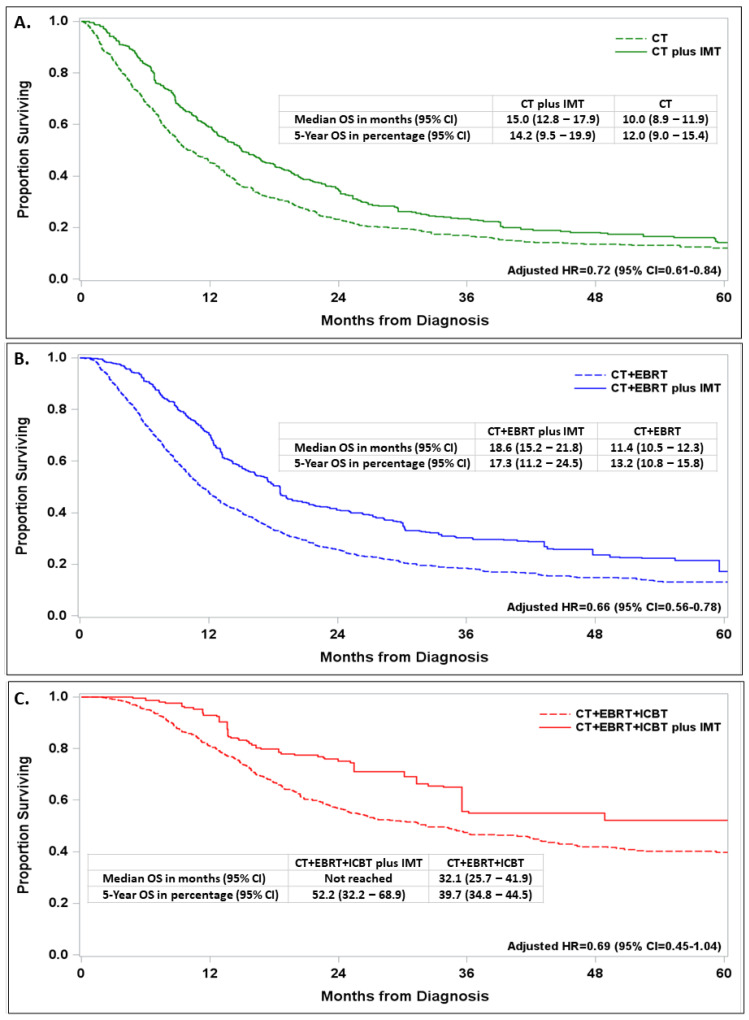
Impact of immune-molecular therapy (IMT) on survival with embedded adjusted hazard ratio (AHR) and 95% confidence interval (CI) in propensity-score-balanced patients diagnosed with stage IVB cervical cancer between 2013 and 2019 and treated with first-line therapy with chemotherapy (CT) alone (**A**), CT and external beam radiation (CT + EBRT) treatment (**B**), or CT + EBRT and intracavitary brachytherapy (ICBT) treatment (**C**).

**Table 1 cancers-16-01071-t001:** Demographic and clinical characteristics of patients with or without the addition of immuno-molecular therapy (IMT) to the first-line therapy in the original cohort and the propensity-score-balanced cohort.

	Total	Original Cohort			Propensity-Score-Balanced Cohort ^i^		
	All Cases	No IMT	IMT ^g^	*p* Value ^h^	No IMT	IMT ^g^	SMD ^j^
	*N* = 3164	*N* = 2195	*N* = 969		*N* = 2103	*N* = 910	
Age at Diagnosis (years)							
[Mean (SD)]	[54.2 (12.7)]	[54.7 (12.9)]	[53.0 (12.1)]	0.0002	[54.2 (12.8)]	[54.4 (12.3)]	2.4%
Race/Ethnicity (%) ^a^				0.005			
Non-Hispanic White	64.2	63.2	68.4		64.1	64.4	0.5%
Non-Hispanic Black	17.7	18.6	15.7		17.9	18.8	2.4%
Hispanic	10.9	11.8	9.1		10.9	10.5	0.2%
Asian/Pacific Islander	4.0	3.8	4.4		4.0	3.8	1.0%
Others/Unknown	3.1	3.5	2.4		3.1	2.6	2.5%
Year of Diagnosis (%)				<0.0001			
2013	11.4	14.8	3.9		11.4	10.7	1.0%
2014	12.8	14.7	8.6		12.9	12.6	0.8%
2015	14.0	15.4	10.8		14.0	13.4	1.8%
2016	15.3	15.4	15.3		15.3	15.6	0.1%
2017	16.2	15.9	16.8		16.0	16.5	0.7%
2018	15.8	12.8	22.7		16.0	16.3	0.2%
2019	14.4	11.2	21.9		14.4	14.9	0.7%
Comorbidity Score (%) ^b^				0.495			
0	80.7	80.6	80.8		80.5	80.0	1.5%
1	13.7	13.5	14.2		13.8	14.3	1.7%
≥2	5.6	5.9	5.0		5.7	5.7	0.0%
Insurance Status (%) ^c^				0.001			
Private Insurance	40.4	38.5	44.8		40.4	39.9	1.8%
Medicare	21.9	22.9	19.5		21.6	21.3	0.8%
Medicaid	26.6	26.6	26.4		26.9	28.5	3.9%
Uninsured	8.7	9.6	6.5		8.6	8.0	1.5%
Unknown	2.5	2.4	2.8		2.5	2.4	0.7%
Facility Type (%) ^d^				0.230			
Academic/Research Program	39.0	38.5	40.4		39.4	40.1	1.3%
Non-Academic/Research	47.3	48.3	45.0		47.2	47.1	0.1%
Other/Unknown	13.7	13.3	14.7		13.5	12.9	1.7%
Neighborhood Income (%) ^e^				0.174			
<USD 40,227	22.3	22.9	21.1		22.2	22.3	0.7%
USD 40,227 to USD 50,353	21.6	21.4	22.0		21.6	21.3	0.8%
USD 50,354 to USD 63,332	19.9	19.3	21.3		19.7	19.0	2.1%
≥USD 63,333	22.4	23.2	20.5		22.4	23.1	1.9%
Unknown	13.8	13.2	15.2		14.1	14.4	0.3%
Geographic Region (%) ^f^				0.027			
Northeast	17.5	17.9	16.6		17.5	16.6	2.5%
Midwest	20.2	19.0	22.9		20.2	20.5	0.2%
South	35.3	36.8	32.1		35.5	37.4	4.0%
West	13.3	13.1	13.7		13.3	12.7	1.4%
Unknown	13.7	13.3	14.7		13.5	12.9	1.7%
Histologic Type (%)				<0.0001			
Squamous Cell Carcinoma	64.7	63.0	68.6		64.5	63.5	2.7%
Adenocarcinoma	16.1	15.5	17.4		16.2	16.3	0.5%
Adenosquamous Carcinoma	3.7	3.0	5.2		3.7	3.7	0.0%
Other Histologic Types	15.5	18.5	8.8		15.6	16.5	4.2%
Tumor Grade (%)				0.441			
1–2	21.2	21.5	20.5		21.2	22.0	1.4%
3	42.5	41.7	44.2		42.4	42.0	0.9%
Not Graded	36.3	36.8	35.3		36.4	36.0	0.3%
Tumor Size (%)				0.387			
<4.0 cm	5.7	5.5	6.2		5.5	5.0	2.6%
4.0–5.9 cm	13.9	13.6	14.6		13.6	13.8	0.2%
6.0–7.9 cm	17.3	17.5	16.7		17.3	17.7	1.2%
≥8.0 cm	17.1	17.9	15.4		17.1	16.7	0.8%
Unknown	46.0	45.5	47.2		46.5	46.9	0.8%
Site of Distant Metastasis (%)				<0.0001			
Distant Lymph Node	26.8	28.4	23.2		26.5	24.2	4.5%
Distant Organ	34.7	34.9	34.4		35.2	36.5	2.1%
Distant Lymph Node and Organ	28.4	25.4	35.0		28.2	29.2	1.6%
Unknown	10.1	11.3	7.4		10.1	10.1	0.9%
First-Line Therapy				<0.0001			
CT	38.1	32.9	49.9		38.0	38.2	0.5%
CT and EBRT	42.7	44.2	39.3		42.8	44.2	2.2%
CT and EBRT and ICBT	19.2	22.8	10.8		19.2	17.6	2.3%

Abbreviations: chemotherapy alone (CT); external beam radiotherapy (EBRT); intracavitary brachytherapy (ICBT); immuno-molecular therapy (IMT). ^a^ Race and ethnicity were defined based on the race and Hispanic ethnicity variables coded by Cancer Registrars in the National Cancer Database (NCDB). ^b^ The comorbidity score in the NCDB was measured using the Charlson–Deyo scoring system and categorized as 0 or ≥1. ^c^ Insurance status in the NCDB was categorized as uninsured, Medicaid, Medicare, or private insurance. Patients with missing insurance data were included. ^d^ Facility type was categorized in the NCDB as Academic/Research, Comprehensive Cancer Center, Integrated Network Cancer Center, Community Cancer, or Other/Unknown. ^e^ Neighborhood income in the NCDB was indicated using the median household income for each patient’s area of residence by matching the zip code against files derived from the 2016 American Community Survey data. Patients with missing neighborhood income data were included. ^f^ Geographic region in the NCDB was categorized as Northeast, Midwest, South, West, Unknown. ^g^ Immuno-molecular therapy treatment (IMT) started to be recorded in the NCDB in 2013 and includes antibody-based therapies such as bevacizumab. ^h^ Difference in demographic and clinical variables across the three groups were compared using *Chi*-square test for categorical variables and an ANOVA test for average age at diagnosis. ^i^ Propensity score analysis was applied using inverse probability of treatment weighting to balance the clinical covariates including age, race/ethnicity, year of diagnosis, comorbidity score, insurance status, facility type, median neighborhood income, geographic region, histology, grade, tumor size, and site of distant metastasis by treatment with CT alone, CT + EBRT, or CT + EBRT + ICBT. ^j^ Standardized mean difference (SMD) was calculated to examine the balance, with a value ≤ 10% considered to be well balanced. *Max* SMD represents the largest SMD observed from the pairwise comparison (no IMT vs. IMT).

**Table 2 cancers-16-01071-t002:** Factors associated with the use of immuno-molecular therapy in patients with stage IVB cervical cancer diagnosed between 2013 and 2019.

Clinical Characteristics	OR (95% CI) ^a^	*p*-Value	Adjusted OR (95% CI) ^b^	*p*-Value
Age				
Each 5-Year Increase	0.91 (0.88–0.95)	<0.0001	0.90 (0.85–0.95)	<0.0001
Insurance Status				
Private	Reference		Reference	
Medicare	0.85 (0.65–1.12)	0.253	0.85 (0.65–1.12)	0.253
Medicaid	0.74 (0.60–0.91)	0.005	0.74 (0.60–0.91)	0.005
Uninsured	0.60 (0.43–0.83)	0.002	0.60 (0.43–0.83)	0.002
Facility Type				
Academic/Research	Reference		Reference	
Non-Academic/Research	0.83 (0.70–0.99)	0.035	0.75 (0.63–0.90)	0.002
Site of Distant Metastasis				
Distant Lymph Node	Reference		Reference	
Distant Organ	0.97 (0.78–1.19)	0.739	0.88 (0.71–1.10)	0.258
Distant Lymph Node and Organ	1.49 (1.20–1.84)	0.0003	1.27 (1.01–1.59)	0.042
First-Line Treatment				
CT	Reference		Reference	
CT and EBRT	0.66 (0.56–0.79)	<0.0001	0.61 (0.51–0.73)	<0.0001
CT and EBRT plus ICBT	0.32 (0.25–0.40)	<0.0001	0.25 (0.19–0.33)	<0.0001

Abbreviations: chemotherapy alone (CT), chemotherapy and external beam radiotherapy (CT and EBRT), chemotherapy and external beam radiotherapy plus intracavitary brachytherapy (CT and EBRT plus ICBT). ^a^ Odds ratio (OR) and 95% confidence interval (CI) of favoring the use of immuno-molecular therapy (IMT) were estimated from unadjusted logistic regression model. ^b^ Clinical factors independently associated with the use of IMT were identified from multivariate analysis using a stepwise logistic regression model, stratified by year of diagnosis and geographic region and adjusted for age, insurance status, facility type, site of distant metastasis, and first-line treatment, with the association expressed using adjusted OR and 95% CI. Race and ethnicity, comorbidity score, neighborhood income, histology, grade, and primary tumor size were less significant and excluded during the stepwise regression.

## Data Availability

Restrictions apply to the dataset: The datasets presented in this article are not readily available. HIPAA-compliant data files for this investigation were acquired on 19 May 2023 through a restricted access approval process from the National Cancer Database with a data use agreement that prohibits sharing (https://www.facs.org/quality-programs/cancer-programs/national-cancer-database accessed on 19 May 2023). Requests to access the datasets should be directed to the following instructions for applications to the National Cancer Database (https://www.facs.org/media/xtvknrsu/2020-puf-instructions-to-potential-applicants.pdf accessed on 19 May 2023).

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
