# Peer review of "Immuno-Molecular Targeted Therapy Use and Survival Benefit in Patients with Stage IVB Cervical Carcinoma in Commission on Cancer®-Accredited Facilities in the United States"

_cancers, 2024, doi:10.3390/cancers16051071_

Round 1

Reviewer 1 Report

Comments and Suggestions for Authors

Immunotherapy use and survival benefit in patients with stage  IVB cervical carcinoma in Commission on Cancer®-accredited  facilities in the United States. By  Collin A et al

is an interesting report based on randomized clinical trials of the benefit associated with immunotherapy in metastatic or recurrent cervical cancer that has spread to other parts of the body (stage IVB ).

The authors in a table (Table 1) intend to describe the clinical characteristics of patients, but only general information and treatment methods are found. Therefore, the lack of certain information, earlier in stage IVB for which the medical profession has decided not to proceed with surgical treatment, may provide crucial information.

This report, although interesting, therefore presents a major gap and cannot be accepted for publication in its current state.

Author Response

See below for our point-by-point response to the requests from Reviewer 1.

REVIEWER 1.

  1. Immunotherapy use and survival benefit in patients with stage IVB cervical carcinoma in Commission on Cancer®-accredited facilities in the United States by Collin A et al. is an interesting report based on randomized clinical trials of the benefit associated with immunotherapy in metastatic or recurrent cervical cancer that has spread to other parts of the body (stage IVB).

RESPONSE: Thank you for your thoughtful review of our original research study. Our study was a retrospective observational investigation using the National Cancer Database (NCDB), which represents 70% of incident cancer diagnoses in the United States (U.S.) across 1,500+ Commission on Cancer (COC)®-accredited facilities, to provide a hospital-based cancer registry view of trends in the use of immuno-molecular therapy with chemotherapy ± radiotherapy over the past decade and its impact on survival in patients with stage IVB cervical cancer in clinical care settings outside of randomized clinical trials.

  1. The authors in a table (Table 1) intend to describe the clinical characteristics of patients, but only general information and treatment methods are found. Therefore, the lack of certain information, earlier in stage IVB for which the medical profession has decided not to proceed with surgical treatment, may provide crucial information. This report, although interesting, therefore presents a major gap and cannot be accepted for publication in its current state.

RESPONSE: Thank you for your feedback. We have modified the limitations section of the Discussion to include the following statement to address this important major gap. “The NCDB provided general characteristics and first-line treatment information for analysis but lacked additional details regarding the patient’s medical, screening, treatment, and surveillance histories to support more in-depth analyses.”

We also modified an additional sentence in the limitations section as follows: “In addition, this study only focused on patients with stage IVB disease without surgery and did not include resected stage IVB patients or those with locally advanced or metastatic stage III or IVA disease who also deserve attention.”

On behalf of my co-authors, we thank you for your reconsideration of our manuscript for the Special Issue in Cancers entitled "Cervical Cancer: Screening and Treatment in 2024"

Sincerely,

Kathleen M Darcy, PhD

[email protected]; 716-228-5703

Reviewer 2 Report

Comments and Suggestions for Authors

In the present manuscript „Immunotherapy use and survival benefit in patients with stage 2 IVB cervical carcinoma in Commission on Cancer®-accredited 3 facilities in the United States”, Sitler et al. analyze real-world data derived from data of the National Cancer Database to assess the potential benefit of the implementation of “immunotherapy”. Authors thereby retrospectively evaluated datasets of 3164 FIGO IVb patients of which 969 received “immunotherapy” as part of their primary treatment.

The introduction is well-written and reflects up-to-date evidence; methods are clearly described, and the statistical approach appears adequate.  A propensity matching was performed to account for potential selection biases. Baseline characteristics are balanced, and crude survival outcomes are approximately in line as previously reported in the KEYNOTE-826 and GOG-240, considering neither study exactly reflected the present patient cohort. The main outcome (figure 4) is clearly presented and relatable. The discussion has a red thread and contextualizes its findings according but is unnecessarily lengthy and convers topics that were not part of this paper: I would suggest to delete the lines “additional research… until optimal treatment modalities” (lines 362-374) without replacement, as mentioned limitations (such as HPV-status) are already discussed in the limitations section.

In line, the conclusion section is too long and not on point. I’d suggest to break it down to 2-3 lines as it is unnecessary to repeat study results in detail, please adapt the following statements for your conclusions and avoid discussion any potential limitations and lengthy thoughts on future research topics. Please also avoid empty phrases such as “life-saving multi-modality treatments”, as the present manuscript is addressing purely palliative therapy approaches.

e.g.:

“Real-world data underlines that.. “immunotherapy” appears to improve survival when added to first-line treatment of stage IVB cervical cancer overall and in all subgroups except for patients with adenocarcinoma. A survival benefit of IMT was also observed in the subset treated with CT, CT+EBRT or CT+EBRT+ICBT… + one concluding sentence”

In summary, the present manuscript addresses a clinically highly relevant topic and provides a methodologically solid analyses of real-world data, supporting previous observations on the clinical benefit of targeted therapies. However: The by far most important drawback of the present study is given in lines 153-155:

“These included agents such as immune checkpoint inhibitors and biologic response modifiers, and in 2013, the NCDB started recording antibody-based agents like bevacizumab as IMT” (lines 153-155)

In fact, we can hardly interpret present data as the label “immunotherapy” comprises a bunch of totally different agents with different efficacies. This issue is data-related and – as far as I understood – not addressable by authors, as the assessed database does not provide detailed data on the applied agent. That the NCBD labels everything down to bevacizumab matters little, as the term “immunotherapy” is grossly misleading in scientific terms, and readers would most likely assume that authors provide an analysis of the addition of checkpoint inhibitors to “standard” therapies – which they do not. This potpourri of therapies under the label of “immunotherapy” allows only for one solid conclusion: That adding targeted therapies appears to improve treatment outcomes in primary metastasized cervical cancers, and this is hardly surprising. Therefore, I have to stress to either 1) address this data issue and, if available, please provide at least a subgroup analysis of patients having received checkpoint inhibitors, as these results would be most interesting to readers; or 2) clearly label that the present paper does not actually evaluate “immunotherapies” – I would suggest to change the designation to “targeted therapies” from the title down to the conclusions, but this is up to the authors.

Author Response

REVIEWER 2.

  1. In the present manuscript „Immunotherapy use and survival benefit in patients with stage IVB cervical carcinoma in Commission on Cancer®-accredited facilities in the United States”, Sitler et al. analyze real-world data derived from data of the National Cancer Database to assess the potential benefit of the implementation of “immunotherapy”. Authors thereby retrospectively evaluated datasets of 3164 FIGO IVb patients of which 969 received “immunotherapy” as part of their primary treatment.

RESPONSE: Thank you for appreciating the high-level features of our original research study.

  1. The introduction is well-written and reflects up-to-date evidence; methods are clearly described, and the statistical approach appears adequate.  A propensity matching was performed to account for potential selection biases. Baseline characteristics are balanced, and crude survival outcomes are approximately in line as previously reported in the KEYNOTE-826 and GOG-240, considering neither study exactly reflected the present patient cohort. The main outcome (figure 4) is clearly presented and relatable. The discussion has a red thread and contextualizes its findings according but is unnecessarily lengthy and convers topics that were not part of this paper: I would suggest to delete the lines “additional research… until optimal treatment modalities” (lines 362-374) without replacement, as mentioned limitations (such as HPV-status) are already discussed in the limitations section.

RESPONSE: Thank you for the positive assessments of our manuscript and appreciate your recommendation to delete this paragraph from the Discussion. We deleted the entire paragraph.

  1. In line, the conclusion section is too long and not on point. I’d suggest to break it down to 2-3 lines as it is unnecessary to repeat study results in detail, please adapt the following statements for your conclusions and avoid discussion any potential limitations and lengthy thoughts on future research topics. Please also avoid empty phrases such as “life-saving multi-modality treatments”, as the present manuscript is addressing purely palliative therapy approaches. e.g.: “Real-world data underlines that.. “immunotherapy” appears to improve survival when added to first-line treatment of stage IVB cervical cancer overall and in all subgroups except for patients with adenocarcinoma. A survival benefit of IMT was also observed in the subset treated with CT, CT+EBRT or CT+EBRT+ICBT… + one concluding sentence”.

RESPONSE: The conclusion paragraph was truncated and modified following your recommendation. Our conclusion section reads as follows.

“5. Conclusions

The use of immuno-molecular targeted agents, labeled as IMT in the NCDB, increased steadily from 11% in 2013 to 46% in 2019 in stage IVB cervical cancer patients in COC®-accredited facilities. The addition of IMT appears to improve survival when added to first-line treatment of stage IVB cervical cancer overall and in all subgroups including those treated with CT, CT+EBRT or CT+EBRT+ICBT but not in patients with stage IVB adenocarcinoma.” 

  1. In summary, the present manuscript addresses a clinically highly relevant topic and provides a methodologically solid analyses of real-world data, supporting previous observations on the clinical benefit of targeted therapies. However: The by far most important drawback of the present study is given in lines 153-155: “These included agents such as immune checkpoint inhibitors and biologic response modifiers, and in 2013, the NCDB started recording antibody-based agents like bevacizumab as IMT” (lines 153-155). In fact, we can hardly interpret present data as the label “immunotherapy” comprises a bunch of totally different agents with different efficacies. This issue is data-related and – as far as I understood – not addressable by authors, as the assessed database does not provide detailed data on the applied agent. That the NCBD labels everything down to bevacizumab matters little, as the term “immunotherapy” is grossly misleading in scientific terms, and readers would most likely assume that authors provide an analysis of the addition of checkpoint inhibitors to “standard” therapies – which they do not. This potpourri of therapies under the label of “immunotherapy” allows only for one solid conclusion: That adding targeted therapies appears to improve treatment outcomes in primary metastasized cervical cancers, and this is hardly surprising. Therefore, I have to stress to either 1) address this data issue and, if available, please provide at least a subgroup analysis of patients having received checkpoint inhibitors, as these results would be most interesting to readers; or 2) clearly label that the present paper does not actually evaluate “immunotherapies” – I would suggest to change the designation to “targeted therapies” from the title down to the conclusions, but this is up to the authors.

RESPONSE: The manuscript has been revised in the following manner to address this important comment.

  • The title was modified as follows: “Immuno-molecular targeted therapy use and survival benefit in patients with stage IVB cervical carcinoma in Commission on Cancer®-accredited facilities in the United States.”
  • The first sentence in the Summary statement was modified as follows: “Summary: Randomized clinical trials show a survival benefit associated with immuno-molecular therapy (IMT) use in metastatic or recurrent cervical cancer.”
  • The label for panel A in the Graphic Abstract was modified as follows: “ Trends in the Use of Immuno-Molecular Therapy (IMT) in Stage IVB Cervical Cancer.”
  • The first sentence in paragraph 5 of the Introduction was modified as follows: “This study aimed to estimate the use of immuno-molecular therapy (IMT) with CT±RT over the past decade and its impact on survival in patients with stage IVB cervical cancer in clinical care settings outside of RCTs.”
  • The beginning of the third paragraph in Section 2.1 of Methods was modified as follows: “IMT was defined in the NCDB as the first course treatment, consisting of biological or chemical agents that would alter the immune system or change the host's response to tumor cells. These included immuno-molecular targeting agents such as immune checkpoint inhibitors and biologic response modifiers, and in 2013, the NCDB started recording antibody-based agents like bevacizumab as IMT. The CT group …”
  • The last sentence in paragraph 1 of Results was modified as follows: “There were 836 patients excluded from this study, including 53 patients treated with hormonal or other first-line treatment, 117 patients treated with an unspecified number of chemotherapy agents, 4 patients with missing immuno-molecular therapy status and 662 patients who started first-line treatment beyond 60 days after diagnosis.”
  • Figure 1 was modified to replace “missing immunotherapy status” with “missing immune-molecular therapy status” in the third bullet and to replace “with addition of IMT” with “with addition of immune-molecular therapy (IMT)” in the second box in the schema.”
  • Table 1 title was modified as follows: “Table 1. Demographic and Clinical Characteristics for Patients with or without the Addition of Immuno-Molecular Therapy (IMT) to First-Line Therapy in the Original Cohort and the Propensity-Score Balanced Cohort.”
  • The abbreviations footnote in Table 1 was modified as follows: “Abbreviations: Chemotherapy alone (CT); External beam radiotherapy (EBRT); Intracavitary brachytherapy (ICBT); Immuno-molecular therapy (IMT).”
  • The title for section 3.1 was modified as follows: “1 Factors Influencing Immuno-Molecular Therapy Use.”
  • Figure 2 title was modified as follows: “Figure 2. Use of Immuno-Molecular Therapy (IMT) in Patients Diagnosed from 2013 to 2019 with Stage IVB Cervical Cancer in the National Cancer Database.”
  • Table 2 title was modified as follows: “Table 2. Factors Associated with the Use of Immuno-Molecular Therapy in Patients with Stage IVB Cervical Cancer Diagnosed between 2013 to 2019.”
  • Footnote a in Table 2 was modified as follows: “Odds ratio (OR) and 95% confidence interval (CI) in favoring the use of immuno-molecular therapy (IMT) were estimated from unadjusted logistic regression model.”
  • The title for section 3.2 was modified as follows: “2 Relationship between Immuno-Molecular Therapy and Survival in Propensity Score-Balanced Patients.”
  • The title for Figure 3 was modified as follows: “Figure 3. Survival following treatment with or without immuno-molecular therapy (IMT) added to chemotherapy alone or chemotherapy with or without radiation (CT±RT) in propensity score balanced patients diagnosed between 2013-2019 with an embedded adjusted hazard ratio (AHR) and 95% confidence interval (CI) for patients treated with CT±RT plus IMT compared with CT±RT (B). Forest plot for all patients with stage IVB cervical cancer diagnosed between 2013-2019 and subgroups of these patients showing the reduced AHR (95% CI) following first-line treatment with CT±RT plus IMT compared with CT±RT (C).”
  • The title for Figure 4 was modified as follows: “Figure 4. The impact of immune-molecular therapy (IMT) on survival with embedded adjusted hazard ratio (AHR) and 95% confidence interval (CI) in propensity score balanced patients diagnosed with stage IVB cervical between 2013-2019 and treated with first-line therapy with chemotherapy (CT) alone (A), CT and external beam radiation (CT+EBRT) treatment (B) or CT+EBRT and intracavitary brachytherapy (ICBT) treatment (B).”
  • The limitations paragraph in the Discussion was modified to include the following sentences: “The IMT variable utilized by NCDB merges multiple classes of immuno-molecular targeting agents together including immune checkpoint inhibitors, biologic response modifiers, molecular targeting agents like Bevacizumab and antibody drug conjugates making it impossible for us to evaluate the individual impact of these classes of agents independently. Our subset analysis in Figure 3B by year of diagnosis suggests a larger survival benefit trend following the addition of IMT in patients diagnosed between 2013-2015, 2016-2017 and 2018-2019 likely associated with the approved addition of bevacizumab in 2014 and ultimately immune checkpoint inhibitors and antibody drug conjugates starting in 2018 for the treatment of patients with metastatic and recurrent cervical cancer.”
  • The conclusions section of the Discussion was modified as follows: “The use of immuno-molecular targeted agents, labeled as IMT in the NCDB, increased steadily from 11% in 2013 to 46% in 2019 in stage IVB cervical cancer patients in COC®-accredited facilities. The addition of IMT appears to improve survival when added to first-line treatment of stage IVB cervical cancer overall and in all subgroups including those treated with CT, CT+EBRT or CT+EBRT+ICBT but not in patients with stage IVB adenocarcinoma.”

On behalf of my co-authors, we thank you for your reconsideration of our manuscript for the Special Issue in Cancers entitled "Cervical Cancer: Screening and Treatment in 2024"

Sincerely,

Kathleen M Darcy, PhD

[email protected]; 716-228-5703

Round 2

Reviewer 1 Report

Comments and Suggestions for Authors

Immuno-molecular targeted therapy use and survival benefit in patients with stage IVB cervical carcinoma in Commission on Cancer®-accredited facilities in the United States. By Collin A. and colleagues, has been revised and currently the objective of the work is more compressible and the missed elements are well discussed. Therefore this report at its current state can be accepted for publication

Reviewer 2 Report

Comments and Suggestions for Authors

The authors answered all my remarks, addressed potential ambiguities accordingly and provided a greatly improved version of the manuscript. I have no further questions to raise.